Loading history changes the morphology and compressive force-induced expression of receptor activator of nuclear factor kappa B ligand/osteoprotegerin in MLO-Y4 osteocytes

http://orcid.org/0000-0003-1884-8053 Wang Ziyi 1 2
Weng Yao 3
http://orcid.org/0000-0002-5999-8253 Ishihara Yoshihito 1
Odagaki Naoya 1
Ei Hsu Hlaing Ei 1
Izawa Takashi 1
http://orcid.org/0000-0002-3254-5238 Okamura Hirohiko 3
Kamioka Hiroshi 1 kamioka@md.okayama-u.ac.jp
1 Department of Orthodontics, Okayama University Graduate School of Medicine, Dentistry, and Pharmaceutical Sciences, Okayama University , Okayama , Japan
2 Research Fellow of Japan Society for the Promotion of Science , Tokyo , Japan
3 Department of Oral Morphology, Okayama University Graduate School of Medicine, Dentistry, and Pharmaceutical Sciences, Okayama University , Okayama , Japan
Stochaj Ursula
Electronic publication date: 2020 Nov 9
Publication date: 2020
Volume: 8
Electronic Location ID: e10244
Received 2020 May 12; Accepted 2020 Oct 5
Copyright: © 2020 Wang et al.
Copyright year: 2020
Copyright holder: Wang et al.
License: This is an open access article distributed under the terms of the Creative Commons Attribution License, which permits unrestricted use, distribution, reproduction and adaptation in any medium and for any purpose provided that it is properly attributed. For attribution, the original author(s), title, publication source (PeerJ) and either DOI or URL of the article must be cited.
License URL: https://creativecommons.org/licenses/by/4.0/

Keywords: Osteocytes, Habituation, 18α-Glycyrrhetinic acid, Fluorescence recovery after photobleaching, Gap junctional intercellular communication

Funding: Grant-in-Aid for Scientific Research 19J11906, 18H03011, 18KK0464 and 19H03859 This work was supported by a Grant-in-Aid for Scientific Research (to Ziyi Wang (19J11906), Takashi Izawa (18H03011, 18KK0464) and Hiroshi Kamioka (19H03859)) from the Japan Society for the Promotion of Science, Japan. The funders had no role in study design, data collection and analysis, decision to publish, or preparation of the manuscript.

==============================
Background

In this study, we investigated the effect of the mechanical loading history on the expression of receptor activator of nuclear factor kappa B ligand (RANKL) and osteoprotegerin (OPG) in MLO-Y4 osteocyte-like cells.

Methods

Three hours after MLO-Y4 osteocytes were seeded, a continuous compressive force (CCF) of 31 dynes/cm2 with or without additional CCF (32 dynes/cm2) was loaded onto the osteocytes. After 36 h, the additional CCF (loading history) was removed for a recovery period of 10 h. The expression of RANKL, OPG, RANKL/OPG ratio, cell numbers, viability and morphology were time-dependently examined at 0, 3, 6 and 10 h. Then, the same additional CCF was applied again for 1 h to all osteocytes with or without the gap junction inhibitor to examine the expression of RANKL, OPG, the RANKL/OPG ratio and other genes that essential to characterize the phenotype of MLO-Y4 cells. Fluorescence recovery after photobleaching technique was also applied to test the differences of gap-junctional intercellular communications (GJIC) among MLO-Y4 cells.

Results

The expression of RANKL and OPG by MLO-Y4 osteocytes without a loading history was dramatically decreased and increased, respectively, in response to the 1-h loading of additional weight. However, the expression of RANKL, OPG and the RANKL/OPG ratio were maintained at the same level as in the control group in the MLO-Y4 osteocytes with a loading history but without gap junction inhibitor treatment. Treatment of loading history significantly changed the capacity of GJIC and protein expression of connexin 43 (Cx43) but not the mRNA expression of Cx43. No significant difference was observed in the cell number or viability between the MLO-Y4 osteocyte-like cells with and without a loading history or among different time checkpoints during the recovery period. The cell morphology showed significant changes and was correlated with the expression of OPG, Gja1 and Dmp1 during the recovery period.

Conclusion

Our findings indicated that the compressive force-induced changes in the RANKL/OPG expression could be habituated within at least 11 h by 36-h CCF exposure. GJIC and cell morphology may play roles in response to loading history in MLO-Y4 osteocyte-like cells.

Introduction

Osteocytes are the most abundant (90–95% of total bone cells in the adult skeleton) and long-lived cell type in bone, which are major regulators of bone mechanosensation and mechanotransduction (Wang et al., 2019; Qin et al., 2020). It has been proposed that the bone cell network stores its mechanical loading history, allowing it to adjust its sensitivity to additional mechanical loading or strain (Turner et al., 2002).

The loading history reshapes the morphology of trabecular bone and the extracellular matrix surrounding osteocytes, as is described in the well-known Wolff’s law (Kerschnitzki et al., 2013), which is also known as bone adaptation. The morphology of the extracellular matrix surrounding osteocytes can influence the extracellular fluid-flow shear stress, thereby enhancing the influence of the loading history on the bone (Kamioka et al., 2012). Consequently, the osteocyte networks are acclimated to daily mechanical stimuli in which high-magnitude strain occurs rarely, and low-magnitude signals occur much more often to accommodate a daily mechanical loading environment (Fritton, McLeod & Rubin, 2000). Previous observations showed that extending the loading duration has a diminishing effect on further bone adaptation, and accommodating to a mechanical loading environment in bone cells makes them less responsive to routine or customary loading signals (Turner & Pavalko, 1998; Robling, 2012).

Xenopus cardiomyocytes and epidermal cell progenitors can be designed by an evolutionary algorithm in silico and engineered to move in desired patterns, such as drawing a circle or writing English letters (Kriegman et al., 2020). However, the lacuna-canalicular system and bone cell network are more complicated than these artificial organisms. Therefore, the bone cell network is considered to be encoded with a unique system for maintaining the history of mechanical stimuli received (Fritton, McLeod & Rubin, 2000; Turner et al., 2002; Moorer & Stains, 2017).

Accumulated evidence has also suggested an additional important role of osteocytes in mobilizing minerals from the surrounding bone matrix via perilacunar/canalicular remodeling (Qing et al., 2012; Lotinun et al., 2019). Therefore, osteocytes may be the principal regulator for the functional bone adaptation (Skedros, Hunt & Bloebaum, 2004; Hazenberg, Lee & Taylor, 2006). Under culture conditions, osteocytes may retain information about their skeletal site of origin. For example, calvarial bone cells are much less sensitive to mechanical stimuli than ulnar bone cells (Rawlinson et al., 2009). Repetitive mechanical loads rapidly desensitize bone cells (Turner, 1998), indicating that bone tissue can be habituated, reducing its response to repetitive mechanical stimuli. Habituation is a term that describes the decrement in responsiveness to a repetitive stimulus in neuronal systems (McDiarmid, Yu & Rankin, 2019).

Osteocytes play a crucial role in bone remodeling by controlling osteoblasts and osteoclasts via the expression of receptor activator of nuclear factor kappa B ligand (RANKL) and its decoy receptor, osteoprotegerin (OPG) (Nakashima et al., 2011; Xiong et al., 2011). Recent studies have shown that the main source of RANKL is osteocytes (Xiong et al., 2015). Correct osteoclastogenesis relies on a correct RANKL/OPG ratio (Capulli, Paone & Rucci, 2014), and osteocytes express both factors at levels comparable with or exceeding those of osteoblasts (Bonewald, 2011).

Based on these findings, we hypothesized that osteocytes might be able to encode their loading history, with this encoded loading history regulating their gene expression even under in vitro culture. It has been stated that, in general, the disuse load was under 200 microstrains, the physiological load was between 200 and 2,500 microstrains, and the overuse loading range was 2,500–5,000 microstrains when applying mechanical loads on the skeleton (Duncan & Turner, 1995; Verbruggen, Vaughan & McNamara, 2012). While, the osteocytes receive more mechanical force since the findings suggested that osteocytes have mechanical signal amplification systems (Takano-Yamamoto, 2014). The application of 2,000 microstrains macroscopically to a piece of bone resulted in a much greater microscopic strain surrounding the osteocyte lacunae of over 30,000 microstrains (Nicolella et al., 2006). A past study used finite element modeling analysis which reported that global compressive loads of 150 microstrains (disuse), 1,000 microstrains (physiological), 3,000 microstrains (overuse) and 5,000 microstrains (pathological overload) resulted in the maximum principal strains of 633, 4,272, 12,820 and 21,528 microstrains respectively (Wang, Dong & Xian, 2018). A previous study already measured the Young’s modulus of MLO-Y4 osteocyte-like cells as 1.98 ± 0.25 kPa and changed very slightly at a variant indentation range (500–1,000 nm) by using atomic force microscopy (Wu et al., 2017). Therefore, we applied approximate 3.1 Pa (resulted in 3.1 Pa/1.98 kPa ≈ 1,566 microstrains) and 6.3 Pa (resulted in 6.3 Pa/1.98 kPa ≈ 3,182 microstrains) compressive force to MLO-Y4 osteocyte-like cells to simulate the bending loads-caused compressive stress in vivo within the pathological load range.

In the present study, MLO-Y4 osteocyte-like cells were proliferated under a continuous compressive force (≈3.1 Pa or 31 dynes/cm2) throughout the entire experiment after seeding the cells 3 h. A long-duration (36 h) additional continuous compressive force (CCF) at 3.2 Pa was applied as the treatment of loading history, followed by re-applying the same additional CCF (≈3.2 Pa or 32 dynes/cm2) for a short duration (1 h), to assess the effect of loading history on the expression of interesting genes as well as the cell number, viability, morphology and cell-to-cell communications of MLO-Y4 osteocyte-like cells.

Materials and Methods

Cell culture and reagents

The MLO-Y4 cell line was purchased from Kerafast (Boston, MA, USA). MLO-Y4 cells (at 39 passages), an osteocyte-like cell line derived from the long bone of a transgenic female mouse containing the osteocalcin promoter driving SV40 T-antigen (Kato et al., 1997), were seeded onto the type I collagen-coated 24-well plates and cultured in alpha-modified Eagle’s minimal essential medium (α-MEM; Thermo Fisher Scientific, Waltham, MA, USA) containing 5.0% heat-inactivated fetal bovine serum (HIFBS; HyClone Laboratories, Logan, UT, USA), 5.0% fetal calf serum (FCS; HyClone, Logan, UT, USA), 100 U/ml penicillin and 100 mg/ml streptomycin (Thermo Fisher Scientific, Waltham, MA, USA) at 37 °C with 5% CO2. MLO-Y4 cells express a high level of connexin 43 (Cx43), early osteocyte markers, such as podoplanin (Pdpn, also know at E11), Osteocalcin (OCN), and low levels of some mature osteocyte markers, such as sclerostin (Sost) and dentin matrix protein 1 (Dmp1) (Yang et al., 2009; Dallas, Prideaux & Bonewald, 2013; Sato et al., 2017).

The gap junction inhibitor 18α-glycyrrhetinic acid (18α-GA) and dimethyl sulfoxide (DMSO) were purchased from Sigma–Aldrich (St. Louis, MO, USA). The following antibodies were used: Rabbit anti-Cx43 polyclonal antibody (Cell Signaling, #3512), Rabbit anti-phospho-Cx43 (Ser368; pCx43) polyclonal antibody (Cell Signaling, #3511), Goat anti-Sost polyclonal antibody (R&D system, AF1589), Mouse anti-β-actin monoclonal antibody (A5441; Sigma–Aldrich, St. Louis, MO, USA), HRP-linked Goat anti-Rabbit IgG (Cell Signaling, #7074S), HRP-linked Rabbit anti-Goat IgG polyclonal antibody (R&D system, HAF017), HRP-linked Goat anti-Mouse IgG polyclonal antibody (AP124P; Millipore, Burlington, MA, USA), Alexa Fluor® 594 phalloidin (A12381; ThermoFisher, Waltham, MA, USA) and Alexa Fluor® 488 linked Goat anti-Rabbit polyclonal antibody (A11008; ThermoFisher, Waltham, MA, USA) used for Western blotting or immunofluorescence staining with MLO-Y4 osteocyte-like cells.

Habituation experiment with CCF

The entire experiment design is shown in Fig. 1. MLO-Y4 cells were seeded onto the type I collagen-coated 24-well plate, 60-mm culture dish, or 35-mm glass-bottom (glass diameter: 14 mm) plastic dish at 2.63 × 104 cells/cm2. A total of 3 h after seeding the MLO-Y4 cells, a round micro-cover glass (Fig. 1A; MATSUNAMI, Japan; area, 78.54 mm2; height, 0.21 mm; diameter, 10 mm; volume, 16.49 mm3; weight, 0.041 g) or an assembly of two pieces of the normal cover glass (Fig. 1A; MATSUNAMI, Japan; area, 960 mm2; height, 0.14 mm; volume, 202.16 mm3; weight, 0.505 g) was placed onto the MLO-Y4 cells as the background CCF of 31 dynes/cm2. As additional CCF, a modified 200-µl pipette tip (Fig. 1A; density, 0.92 g/cm3; height, 16 mm; volume, 28.26 mm3; weight, 0.211 g) was placed onto the micro-cover glass. The modified 200-µl pipette tip itself generated 32 dynes/cm2 CCF when it was immersed into the well of the 24-well plate with 2.1 ml culture media. Therefore, as shown in Fig. 1, the MLO-Y4 cells in the loading history group were grown under a total of 63 dynes/cm2 CCF, while the cells in the control group were grown under only the background CCF (31 dynes/cm2).

Figure 1 Overview of the methods in this study.

(A) The continuous compressive system and (B) the overview of the experiment design. In the present study, MLO-Y4 osteocyte-like cells were proliferated under a continuous compressive force (≈3.1 Pa or 31 dynes/cm2) throughout the entire experiment after seeding the cells 3 h. A long-duration (36 h) additional continuous compressive force (CCF) at 3.2 Pa was applied as the treatment of loading history, followed by re-applying the same additional CCF (≈3.2 Pa or 32 dynes/cm2) for a short duration (1 h), to assess the effect of loading history on the expression of interesting genes as well as the cell number, viability, morphology, and cell-to-cell communications of MLO-Y4 osteocyte-like cells.

After maintaining the MLO-Y4 cells under additional CCF for 36 h, the additional CCF was removed for 10 h (recovery period) before the same additional CCF was applied again for another hour. One hour before applying the additional CCF again for an extra hour, 3.0 µM 18α-GA or 0.1% DMSO was added and kept in the medium until the end of this experiment (Fig. 1B).

Calcein-acetoxymethyl ester and Hoechst 33342 staining

During the 10-h recovery period, the cell number and viability were examined at 0, 3, 6 and 10 h using Calcein-acetoxymethyl (Calcein-AM) and Hoechst 33342 staining. The micro-cover glasses were removed 15 min prior to each time checkpoint, and the cells were loaded with 1.25 µM Calcein-AM and 1 µg/ml Hoechst 33342 in the culture media for 15 min. Fluorescent and phase-contrast images of the area that had been under the micro-cover glass were then taken with a KEYENCE BZ-9000 fluorescence microscope (KEYENCE, Osaka, Japan) using a 20x phase-contrast lens.

Cell numbers, viability and morphological measurements

After routine background correction using the ImageJ/Fiji software program (Schindelin et al., 2012), the background was neutralized using the “subtract background” (Sternberg, 1983) feature of the ImageJ/Fiji tool. Finally, the number of cells with positive staining for Calcein-AM and Hoechst 33342 and the cell morphological results were analyzed using the “Analyze Particles” function after auto-thresholding with the default method in the ImageJ/Fiji tool.

Cell numbers were defined as the number of cell nuclei with positive staining on Hoechst 33342. The viable cell number was defined as the number of cells with positive staining on both Calcein-AM or Hoechst 33342.

For the cell morphology, the aspect ratio, circularity, and solidity were measured. The aspect ratio was defined as the ratio of the major axis to the minor axis of the best-fitted ellipse. Circularity was defined as the 4π× cell area/perimeter2. Solidity was defined as the ratio of the cell area to the convex area.

Reverse transcription and quantitative real-time polymerase chain reaction

Total RNA was extracted using ISOGEN (Nippon Gene, Tokyo, Japan) and was used to synthesize complementary DNA (cDNA) with a ReverTra Ace quantitative real-time polymerase chain reaction (qRT-PCR) Kit (FSQ-201; Toyobo Co., Ltd., Osaka, Japan) in the total RNA concentration of 100 ng/µl. The resulting cDNA products were diluted five times using pure water, and 1 µl of five times diluted-cDNA product was used as a template to quantify the relative content of messenger RNA (mRNA) by a qRT-PCR using SYBR® Green Real-time PCR Master Mix (QPK-201; Toyobo Co., Ltd., Osaka, Japan). The relative levels of the PCR products were determined using a LightCycler System (Roche Diagnostics, Mannheim, Germany). The values of the threshold cycle (Ct) were determined automatically by LightCycler 96 software (version 1.1; Roche Diagnostics, Mannheim, Germany) with the default setting. The sequences of the primers used in this study can be downloaded from Mendeley Data (http://dx.doi.org/10.17632/2yfd2w8jfp.1#file-c35071d0-dec9-47c0-84dc-351288a8356c). Differences in gene expression levels following treatment were calculated using the 2−ΔΔCt method after normalization within each sample of interesting gene expression levels against the expression levels of the reference genes (Gapdh).

Fluorescence recovery after photobleaching assay and immunofluorescence staining

The MLO-Y4 osteocyte-like cells cultured in the glass-bottom dish were loaded with 1.25 µM Calcein-AM 1 h prior to the time point (45 h in Fig. 1B) of re-applying CCF. The culture medium was refreshed two hours after loading with Calcein-AM, then the Fluorescence recovery after photobleaching (FRAP) assay was performed alternately among the normal group (without cover glass and pipette tip), control group (cultured with glass only), loading history group (cultured with glass and pipette tip after loading history treatment), and CCF group (cultured with glass and pipette without loading history treatment) within 90 minutes.

FLUOVIEW FV500 confocal laser scanning microscopy (CLS) system (Olympus, Tokyo, Japan) equipped for differential interference contrast (DIC) microscopy. The CLS microscopy system was coupled to an inverted microscope (IX-71; Olympus) with a ×60 (N.A. = 1.4) oil-immersion objective lens. The scanning rate was 1.66 s/scan for a 16-bit image, 512 × 512 pixels in size. An MLO-Y4 cell surrounded by other cells under the cover glass (if applied), which was confirmed by the DIC view, was chosen for photobleaching. The boundary was enclosed and outlined with a rectangular region-of-interest tool. A predefined three-step FRAP procedure was automatically executed at a scan speed of 1.66 s/scan on the largest cross section of the target cell. In the first step, a prebleached image of the whole field was taken using a low laser intensity (AOTF = 10%, zoom = ×1). The laser intensity was then increased ×100 (AOTF = 100%, zoom = ×40), and the target cell was photobleached one time for about 2–3 s (depending on the area of the target cell). Finally, the laser intensity and zoom were immediately reset to the prebleach levels (AOTF = 10%, zoom = ×1), and time-lapse images were acquired with an interval of 15 s.

The average intensity at each imaging time point was measured for three regions of interest: the bleached target cell (It), all other cells in the image field (Tt) as control, and the non-fluorescent region outside of all of the cells for background subtraction (BG) using the “Time Series Analyzer” tool of ImageJ software (https://imagej.nih.gov/ij/plugins/time-series.html). The fluorescence intensity of the target osteocyte (F) was normalized as follows (Phair, Gorski & Misteli, 2003): (1) Ft=(Tprebleach−BG)(It−BG)(Tt−BG)(Iprebleach−BG)

The replacement of fluorescence within a bleached cell (Rt) was calculated using the following equation (Ishihara et al., 2008; Wang et al., 2016): (2) Rt=[(Ft−F0)/(Fi−F0)]×100(%)

The percent replacement was defined as the fraction of molecules that were replaced during the time-course of the experiment. Ft is the normalized fluorescence intensity at the time (t) after photobleaching by Eq. (1). F0 is the normalized fluorescence immediately after photobleaching. Fi is the initial fluorescence intensity before photobleaching.

Recovery curves could be analyzed for passive transport of fluorescent dyes through MLO-Y4 osteocyte-like dendritic processes connected by gap junctions. Its kinetics follow the equation (Wade, Trosko & Schindler, 1986): (3) Rp−RtRp−R0=e−kt

where Rp, R0, and Rt, are replacement of fluorescence in the bleached target cell at plateau (equilibrium), zero time and time (t) after bleaching, respectively. The Eq. (3) could be rearranged as Rt=R0+(Rp−R0)×(1−e−kt), which is also called one-phase exponential association equation. The parameter of k and Rp were estimated by a curve fit in Graphpad Prism software (GraphPad Software, Inc., San Diego, CA, USA).

If we ignore the replacement of fluorescence during the photobleaching period, we could take the Rp as the mobile fraction (fm). The rate coefficient k is positively related to the permeability coefficient of calcein.

To verify the protein expression of Cx43 and gap-junctional connections between MLO-Y4 cells, an immunofluorescence staining was performed. MLO-Y4 osteocyte-like cells on a glass-bottom dish were fixed with 4% paraformaldehyde in phosphate-buffered saline (PBS) for 10 min, then permeabilized by incubation in 0.5% Triton X-100 in PBS for 10 min. The cells were blocked with Blocking One Histo (Nacalai tesque, Osaka, Japan) for 8 min at RT. After blocking, the cells were incubated with Rabbit anti-Cx43 polyclonal antibody (overnight at 4 °C), Alexa Fluor® 488 linked Goat anti-Rabbit polyclonal antibody (1 h at RT), Alexa Fluor® 594 phalloidin (1 h at RT), and Hoechst 33342 (10 min at RT) in PBST containing 1% bovine serum albumin (BSA; Sigma–Aldrich, St. Louis, MO, USA). The cells were rinsed three times with PBST after each above-mentioned step.

Western blot analyses

After the above-mentioned habituation experiment with CCF, the cells were washed with cold PBS and lysed with a lysate buffer (1 mM Dithiothreitol, 1 mM Phenylmethylsulfonyl Fluoride, 1 μg/mL leupeptin, 2 μg/mL aprotinin, 5 mM EGTA) and sonicated on ice using a supersonic machine. The protein concentration was determined using the Pierce™ BCA Protein Assay Kit (Thermo Scientific, Waltham, MA, USA) and adjusted to a concentration of 1.1 mg/ml with the lysate buffer. Fifteen micrograms of each sample and Precision™ Plus Protein Dual Xtra Standards markers (Bio-Rad, Berkeley, CA, USA) were separated by 7.5% or Any kD™ Criterion™ TGX™ Precast Gel (Bio-Rad, Berkeley, CA, USA) according to the predicted molecular weight (MW) of our interesting proteins. The separated proteins were then transferred to PVDF membranes (Millipore, Billerica, MA, USA). The membranes were blocked in a solution consisting of 5% nonfat skim milk in PBS containing 0.1% Tween-20 (PBST) for 1 h, and then incubated 1 h at room temperature (RT) in Can Get Signal Immunoreaction Enhancer solution (NKB-101; Toyobo Co., Ltd., Osaka, Japan) containing specific antibodies. The membranes were washed three times (10 min each time) in PBST, then incubated for 1 h at RT with the corresponding secondary antibodies. After washing three times, as described above, the proteins were detected by a chemiluminescence system (ChemiDoc™ XRS+; Bio-Rad, Berkeley, CA, USA) with 20X LumiGLO® Reagent and 20X Peroxide (Cell Signaling, Danvers, MA, USA). The densitometric analysis of bands was performed using the ImageLab software (version 6.0.1; Bio-Rad, Berkeley, CA, USA). The MW of our interesting proteins was estimated by a standard curve of the logarithm of the MW versus relative migration distance that was generated using the Precision™ Plus Protein Dual Xtra Standards.

Statistical analyses

All experiments in this study were performed in biological triplicated or quadruplicated wells or dishes with different batches. The cell that was cultured in 24-well plate within the first batch were used for morphological measurements and RNA extraction (see the assignment of treatment in Fig. S1). The cells cultured in 60 mm plastic dishes in the second batch were used for protein extraction. The cells cultured in glass-bottom plastic dishes in the last batch were used for FRAP experiments and immunofluorescence staining. The normality of all results was tested by the Shapiro–Wilk test. A one-way analysis of variance (ANOVA) followed by a multiple comparisons test with Fisher’s least significant difference (LSD) test or with false discovery rate (FDR) control was performed for the results with a normal distribution, while the Kruskal-Wallis test followed by a multiple comparisons test with FDR control was performed for the results with a non-normal distribution. The correlation of the expression of interesting genes with the cell morphology (aspect ratio, circularity and solidity) was examined by Pearson’s correlation coefficient and linear regression, but only significant correlations were plotted out in this study. A two-way ANOVA followed by a multiple comparisons test with FDR control was applied to test the differences of intracellular fluorescence intensity of calcein among different treatments at each time checkpoint during the FRAP experiments.

All of the statistical analyses were performed using the GraphPad Prism software program, version 8.0.0 for Windows (GraphPad Software, Inc., San Diego, CA, USA).

Results

MLO-Y4 cells maintained high viability under both 31 and 63 dynes/cm2 for at least 36 h

As Figs. 2A and 2B show, the MLO-Y4 cells survived under both 31 and 63 dynes/cm2 CCF for at least 36 h. No significant differences in the cell number (239 cells/mm2 on average in Fig. 3A) or viability (93.67% on average in Fig. 3B) under the micro-cover glass were observed between the MLO-Y4 osteocytes with and without a loading history or among different time checkpoints during the recovery period.

Figure 2 Calcein-AM and Hoechst 33342 staining and morphological measurements.

(A) An example of the phase-contrast image with Calcein-AM and Hoechst 33342 staining. (B) The morphological changes during the recover-period. (C) An illustration of the morphological measurements. The right samples experienced an additional 32 dynes/cm2 loading before this 10 h recover-period as the treatment of loading history comparing to the left samples. All original images used for the morphological measurements can be downloaded from Mendeley Data (http://dx.doi.org/10.17632/2yfd2w8jfp.1#folder-36eef903-4de0-4fcc-916a-b27262fb31a1).

Figure 3 Proliferation of MLO-Y4 osteocytes during recovery period.

The results of cell number counting (A) and cell viability (B). ANOVA, analysis of variance; the P-values for each ANOVA test were listed before the legends; the P-values for each time point were from the independent Student’s t-test. The values of each replicate are presented as the dots. The results are shown as mean ± standard deviation.

The morphology changed during the recovery period

Figure 2C shows an example of the morphological measurements. At the end of this 10 h recover-period, the aspect ratio was significantly increased only in the MLO-Y4 osteocytes without a loading history (median aspect ratio of 1.7 vs. 2.0 with FDR value less than 0.001 in Fig. 4A). The circularity of MLO-Y4 osteocytes with a loading history was lower than that of the cells without a loading history immediately after removing the additional weight but ultimately returned to the same level after the 10-h recovery period (Fig. 4B). The solidity (cell area / convex area) of the MLO-Y4 osteocytes with a loading history was lower than that of the cells without a loading history after the 10-h recovery period but the same level as when immediately removing the additional weight (Fig. 4C). Similarly to the solidity, the cell area did not show any significant differences between the MLO-Y4 cells under 63 dynes/cm2 and 31 dynes/cm2 CCF for 36 h, although the cell area was significantly different after the 10-h recovery period (Fig. 4D).

Figure 4 The results of morphological measurements during the recovery period.

The aspect ratio (A), circularity (B), solidity (C), and cell area (D) are shown in a violin plot. The results of each cell are represented as the dots inside the violin shape; the kernel density is represented as the outline of the violin shape; the red solid line indicates the median value; the red dashed line indicates the quartiles. FDR, false discovery rate; the FDR test was performed following the Kruskal-Wallis test. All original images used for cell morphological measurements can be downloaded from Mendeley Data (http://dx.doi.org/10.17632/2yfd2w8jfp.1#folder-36eef903-4de0-4fcc-916a-b27262fb31a1).

The expression of most of the expression of our interesting genes recovered to the control level after removal of the mechanical stimuli

The mRNA expression of RANKL, OPG, Gja1, and Pdpn was decreased, and the RANKL/OPG ratio, Sost, Dmp1, and OCN higher in the MLO-Y4 osteocyte-like cells under 63 dynes/cm2 than in those under 31 dynes/cm2 (Fig. 5). After changing the CCF from 63 dynes/cm2 back to 31 dynes/cm2 for 3 h, the mRNA expression of RANKL, Gja1, Dmp1, Pdpn, and OCN recovered back to the same level as in the MLO-Y4 osteocyte-like cells under 31 dynes/cm2 (Figs. 5A, 5D, 5F, 5G and 5H), but the mRNA expression of OPG increased, so the RANKL/OPG ratio consequently decreased (Fig. 5C). Ultimately, the expression of RANKL and OPG and the RANKL/OPG ratio returned to the same level as in the control group after changing the CCF from 63 dynes/cm2 back to 31 dynes/cm2 for 10 h (Figs. 5A–5C). However, the mRNA expression of Sost still higher in the cells previously under 63 dynes/cm2 than in those only under 31 dynes/cm2 even after changing the CCF from 63 dynes/cm2 back to 31 dynes/cm2 for 10 h (Fig. 5E). Notably, the mRNA expression of Dmp1 was suddenly increased in the cells previously under 63 dynes/cm2 (with loading history) at the end of this 10-h recovery period (Fig. 5F).

Figure 5 Expression profile of interesting genes and correlated morphological changes during the recovery period.

The mRNA expression of (A) RANKL, (B) OPG, (C) RANKL/OPG, (D) Gja1, (E) Sost, (F) Dmp1, (G) PDNP and (H) OCN during the recovery period and the correlations of (I) OPG with Circularity, (J) of Gja1 with Circularity and (L) of Dmp1 and Area. Independent Student’s t-test was performed to test the differences of the mRNA expression of interesting genes between MLO-Y4 with and without loading history at each time point during the recovery period. *P < 0.05; **P < 0.01; ***P < 0.001. The dash lines in (I), (J), (K) and (L) indicate the 95% confidence interval. The values of each replicate are presented as the dots. The results are shown as mean ± standard deviation.

The expression profiles of OPG, Gja1 and Dmp1 were significantly correlated with the cell morphological changes during the recovery period

The mRNA expression profile of OPG, Gja1 and Dmp1 was significantly correlated with the morphological changes in circularity, solidity, or area during the 10-h recovery period (Figs. 5I–5L). Interestingly, more significant changes of all of the solidity (Fig. 4C), area (Fig. 4D), and the mRNA expression profile of Dmp1 (Fig. 5F) were observed at the end of the 10-h recovery period. Please note that the Pearson’s Correlation Coefficient was calculated using the mean value of both morphological changes and mRNA expression fold change, but the linear regression test was performed using the individual mRNA expression fold change against the mean value of morphological changes.

The loading history reduced the responsiveness when the same CCF was applied again but could be influenced by gap junctional inhibitor

The mRNA expression of RANKL, OPG and the RANKL/OPG ratio in the MLO-Y4 osteocytes without a loading history was dramatically changed (i vs. iii in Figs. 6A–6C) in response to the new 1-h application of additional weight loading, but no significant changes were observed in the MLO-Y4 osteocyte-like cells with a loading history (i vs. ii in Figs. 6A–6C). However, the expression of RANKL, OPG and the RANKL/OPG ratio remained at the same level as in the control group in the MLO-Y4 osteocyte-like cells with a loading history (i vs. ii in Figs. 6A–6C), but similar findings were not found in the MLO-Y4 osteocyte-like cells treated with gap junction inhibitor (v vs. viii in Figs. 6A–6C). In other words, the mRNA expression of RANKL, OPG and the RANKL/OPG ratio was habituated by a 36-h CCF and was disrupted by junction inhibitor.

Figure 6 The influences of loading history and 18α-GA on compressive force-induced changes of mRNA expression of interesting genes in MLO-Y4 osteocyte-like cells.

The mRNA expression of (A) RANKL, (B) OPG, (C) RANKL/OPG, (D) Gja1, (E) Sost, (F) Dmp1, (G) PDNP and (H) OCN of MLO-Y4 osteocyte-like cells in response to the new applied or re-applied continuous compressive force (1 h) with or without 18α-GA treatment (2 h). The values of each replicate are presented as the dots. The results are shown as mean ± standard deviation. ANOVA, analysis of variance; Fisher’s least Significant Difference (LSD) test was performed following the ANOVA test. *P < 0.05; **P < 0.01; ***P < 0.001; ****P < 0.0001.

Notably, 18α-GA treatment reduced the RANKL/OPG ratio by decreasing the RANKL expression (iv vs. v in Figs. 6A and 6B). However, in response to the new application of 1-h loading of additional weight in the MLO-Y4 osteocyte-like cells without a loading history (i vs. iii and v vs. vii in Figs. 6A–6C), the mRNA expression of OPG showed a similar tendency to that in the MLO-Y4 osteocytes with 18α-GA (comparing i vs. iii with v vs. vii in Figs. 6B), but the changes in the MLO-Y4 osteocytes with 18α-GA were less significant than the MLO-Y4 osteocytes without a loading history (comparing i vs. iii with v vs. vii in Figs. 6B). Interestingly, the increased RANKL, the decreased OPG, and the increased RANKL/OPG ratio were observed in the 18α-GA-treated MLO-Y4 osteocyte-like cells with a loading history in response to the re-application of 1-h loading of additional weight (v vs. viii in Figs. 6A–6C), a finding that was completely the opposite of that in MLO-Y4 osteocyte-like cells without both 18α-GA treatment and loading history(comparing i vs. iii with v vs. viii in Figs. 6A–6C).

Whereas, the rest of our interesting genes (Figs. 6D–6H) did not show such obvious habituation phenomena or opposite trend between with and without 18α-GA in response to the new 1-h application of additional weight loading as described above. However, the increase of mRNA expression of Sost and OCN in the MLO-Y4 osteocyte-like cells with a loading history is less than that in the MLO-Y4 osteocyte-like cells without a loading history (ii vs. iii in Figs. 6E and 6H). In other words, the mRNA expression of Sost and OCN showed sensitivity to the treatment of loading history (ii vs. iii in Figs. 6E and 6H).

18α-GA blocks the gap-junctional intercellular communications (GJIC), which was shown in our previous studies (Kamioka et al., 2007; Ishihara et al., 2008). However, the mode of inhibitory action of GA is still not completely understood (Willebrords et al., 2017). Direct interaction between 18α-GA and gap junctions (GJs) is possible when the former is inserted into the plasma membrane, thereby binding to GJs and causing a conformational alteration (Willebrords et al., 2017). Other possibilities include changes in the connexin phosphorylation status, which led to a reduction in connexin expression (Willebrords et al., 2017). Our results (iv vs. v in Fig. 6D, P-value < 0.05 by unpaired t-test which did not show) also showed a reduction of Gja1, which were in good agreement with the previous results (Willebrords et al., 2017), although this reduction did not show significance by Fisher’s LSD due to the big pooled variance.

GJIC and Cx43 accumulation was reduced by loading history

The GJIC among the MLO-Y4 osteocyte-like cells was confirmed by the immunofluorescence staining of Cx43 and FRAP assay (Figs. 7A and 7B). Comparing with normal cultured MLO-Y4 cells (i in Figs. 7C–7G and 8), the MLO-Y4 cells cultured under glass (ii in Figs. 7C–7G and 8) with (iii in Figs. 7C–7G and 8) or without (iv in Figs. 7C–7G and 8) loading history showed decreased mobile fraction (i.e., lower recover percentage) in Figs. 7C and 7D with decreased protein expression of Cx43 (Figs. 8A and 8B). Remarkably, the changes in the capacity of GJIC and protein expression of Cx43 were opposite between the MLO-Y4 osteocyte-like cells with and without treatment of loading history (comparing ii vs. iii with ii vs. iv in Figs. 7C, 7D, 8A and 8B) in response to the application of 1-h loading of additional weight. Indeed, the significant interaction effect was observed between mechanical stimuli (i.e., normal cultured, cultured under cover glass with or without loading history) and time after photobleaching in Fig. 7C, which suggested a significant influence of the loading history on the fluorescence recover curve. The MLO-Y4 osteocyte-like cells without loading history showed a sharper recover curve and higher coefficient k than that with loading history. In other words, the loading history disrupted the CCF-caused GJIC increase. High R2 values shown in Fig. 7F suggested a good fitness of the actual recover curve to the theoretically predicted curve (Fig. 7C). No significant differences were observed in the number of neighboring cells, which implied that the GJIC differences among groups in this study were not caused by the different number of neighboring cells.

Figure 7 Cx43 expression and fluorescence recovery after photobleaching (FRAP) assay in MLO-Y4 osteocyte-like cells.

(A) The expression of Cx43 in normal cultured (without cover glass) MLO-Y4 osteocyte-like cells. (B) The instrument scheme and examples for FRAP assay in MLO-Y4 osteocyte-like cells. (C) FRAP recover curve and fitted model of calcein fluorescence transportation in MLO-Y4 osteocyte-like cells with or without loading history. After a two-way ANOVA test, multiple testing with false discovery rate (FDR) control for the recover curves was performed and transformed FDR values are plotted (higher plot bar with lower original FDR value). The (D) predicted mobile fraction, (E) coefficient k and (F) goodness of fit from the fitted model were investigated. (G) The number of neighbouring cells surrounding the photobleached target cell were counted. All original time-lapse images used for FRAP analysis can be downloaded from Mendeley Data (DOI: http://dx.doi.org/10.17632/2yfd2w8jfp.1#folder-177a56ec-cb22-492b-8f72-21438f23394c). In (D), (E), (F) and (G), Fisher’s least Significant Difference (LSD) test was performed following the ANOVA test, the values of each replicate are presented as the dots, and the results are shown as mean ± standard deviation. The results in (C) are shown as the mean ± standard error of the mean. Scale bar = 20 µm. ANOVA, analysis of variance; *P < 0.05; **P < 0.01.

Figure 8 Protein expression changes of Cx43, pCx43, and Sost of MLO-Y4 osteocyte-like cells in response to re-applied compressive force with or without loading history.

(A) Loading history suppressed the mechanosensitivity of Cx43 and pCx43 but not Sost in response to the compressive force applied again. The quantitative results of (A) for protein expression of (B) Cx43, (C) pCx43, (D) pCx43/Cx43 and (E) Sost. Multiple testing with false discovery rate (FDR) control followed a one-way ANOVA test was performed by using the results from two independent replications. The values of each replicate are presented as the dots. The results are shown as mean ± standard deviation. ANOVA, analysis of variance. *FDR < 0.05; **FDR < 0.01; ***FDR < 0.001. The molecular weight (MW) was estimated by a standard curve of the logarithm of the MW versus relative migration distance that was generated using the PrecisionTM Plus Protein Dual Xtra Standards.

The ratio of pCx43 to total Cx43 was unchanged among the different treatment of mechanical stimuli (Figs. 8A and 8D). Although the mRNA expression of Sost showed sensitivity to the treatment of loading history (ii vs. iii in Fig. 6E), the protein expression of Sost did not show significant differences between the MLO-Y4 osteocyte-like cells with and without loading history (ii vs. iii in Figs. 8A and 8E).

Discussion

In this study, the phenotype of MLO-Y4 cells was confirmed by checking the mRNA expression of Pdpn, Cx43, RANKL, OCN, Dmp1, Sost and OPG, respectively, with the average Ct values of 18.13, 19.75, 22.22, 23.74, 28.93, 30.74 and 33.92 comparing to the average Ct = 14.03 of Gapdh (raw Ct values could be downloaded from Mendeley Data: http://dx.doi.org/10.17632/2yfd2w8jfp.1#file-22aa2376-6feb-4709-b37e-dfdc8c8cf309). The protein expression of Cx43 and Sost was confirmed by Western blot (Fig. 8A). The GJIC was confirmed by the FRAP assay and immunofluorescence staining of Cx43 (Fig. 7A and 7B). The dendritic cellular processes were also observed (Figs. 2, 7A and 7B). Therefore, the MLO-Y4 cells used in this study were characterized as the osteocyte-like cells.

Our results showed that MLO-Y4 cells maintained high viability under both 31 and 63 dynes/cm2 for at least 36 h (Fig. 3B); however, the proliferation may be inhibited. The grown area covering each well of the 24-well plate was 1.9 cm2, so the estimated cell number in total was 45,410 cells/well, based on our results shown in Fig. 3A (239 cells/mm2), which was even smaller than our seeding density (50,000 cells/well). The difference in the density between the MLO-Y4 cells grown under a cover glass and without a cover glass was very obvious at the edge of the cover glass (Fig. S2). This may be because placing the cover glass directly on the cell may have created a relatively sealed environment that reduced nutrition. This relatively sealed environment is similar to the environment of mineralized tissue. Moreover, this condition also reduced the proliferation of MLO-Y4 osteocytes that may impact the cell morphology and gene expression profile. A previous microarray study showed that the MLO-Y4 osteocytes had a different gene expression profile between conditions of low and high cell density (Yang et al., 2009). Cell density may also influence cell morphology. A relatively sealed environment combined with the inhibition of proliferation ultimately created an ideal condition for us to observe the cell morphological changes during the 10-h recovery period after 36-h additional CCF exposure.

In the current study, we developed an in vitro platform mimicking the habituation phenomena observed in in vivo or ex vivo bone tissues that provide a more economic and convenient approach to investigating the possible mechanism of encoding the loading history in osteocytes. With this platform, all MLO-Y4 osteocytes were grown under background CCF, which is similar to the in vivo conditions, as all osteocytes in our bodies are growing under a complicated loading environment (Turner, 1998). Unlike to hydrostatic pressure, compressive pressure generates the uniaxial deformation that is also an important mechanical type received by osteocytes in vivo by daily activities (Duncan & Turner, 1995).

In vitro cultured MLO-Y4 osteocytes were successfully habituated

The reduced responsiveness of the RANKL/OPG expression to the re-application of the same CCF was diminished in the MLO-Y4 osteocytes with a loading history, suggesting that the in vitro cultured MLO-Y4 osteocytes could be habituated by 36-h CCF application in a loading environment with background CCF (comparing i vs. iii with i vs. ii in Figs. 5A–5C and 6A–6C). Loading history significantly suppressed the CCF-induced increase of mRNA expression of Sost and OCN (ii vs. iii in Figs. 6E and 6H). However, no significant differences in protein expression changes (iii vs. iv in Figs. 8A and 8E) in Sost between the MLO-Y4 osteocyte-like cells with and without loading history in response to the application of the additional loading weight for 1 h. Treatment of loading history reversed the response of the protein expression of Cx43 and GJIC to the re-application of the same CCF for 1 h (comparing ii vs. iii with ii vs. iv in Figs. 7C, 7D, 8A and 8B). However, the treatment of loading history did not show any significant influences on the mRNA expression of Cx43 (ii vs. iii in Fig. 6D).

It is well documented that mechanical stimuli changed the protein secretion in many other types of cells (Apodaca, 2002). Since the Sost is a secreted protein (Poole et al., 2005), the discrepancy between the mRNA and protein expression of Sost may be due to the mechanical stimuli-induced changes in protein secretion. Indeed, our previous study found that mechanical unloading changed the concentration of extracellular Sost but not the intracellular Sost in the femur in vivo using a transmission electron microscopy (Osumi et al., 2020). On the other hand, the discrepancy between the mRNA and protein expression of Cx43 may be due to the post-translational modifications, since the mechanical stimulation changed the phosphorylation status of Cx43 (Genetos et al., 2007; Qin et al., 2020) and Cx43 degradation is controlled by complex crosstalk between connexin phosphorylation and ubiquitination (Totland et al., 2020). Ser368 of Cx43 is phosphorylated by protein kinase C (PKC), which decreases cell-to-cell communication (Lampe et al., 2000). However, the pCx43/Cx43 ratio in the Figs. 8A and 8D did not show significant differences among all groups, which indicated that the loading history-induced changes in GJIC and protein expression of Cx43 in this study was not due to the phosphorylation of Ser368 of Cx43 by PKC. Mitogen-activated protein kinase (MAPK)-mediated phosphorylation of Cx43 at serine residues 255, 262, 279, and 282 was related to the Cx43 degradation (Totland et al., 2020). Lots of studies showed that the MAPK pathway could be rapidly activated by various cellular mechanical stimuli (Takano-Yamamoto, 2014). Therefore, it is worthy of further studies in the future which the role of the MAPK pathway in the loading history mediated gene expression revealed by this study. The above discussions indicated that loading history might also have an influence on the post-translational modifications on proteins, which is not fully supported by this study but worthy of further investigations.

Blockade of junctions influenced the habituation effect

Gap junctions are hexametric channels formed by two docked hemichannels from adjacent cells that permit the direct intercellular transfer of small signaling molecules, such as inositol phosphates, cyclic nucleotides, and ATP. Connexins are the main components of gap junctions and hemichannels, and the most abundant connexin in bone cells is Cx43 (Ishihara et al., 2008; Manuscript, Plotkin & Bellido, 2013), which is highly expressed in MLO-Y4 cells (Kato et al., 1997). Cx43 is a key component of intracellular machinery responsible for signal transduction in bone in response to variant stimuli (Manuscript, Plotkin & Bellido, 2013). Our results showed that 18α-GA treatment of MLO-Y4 osteocytes with a loading history rescued their sensitivity to RANKL and OPG expression in response to the re-application of CCF (comparing i vs. ii with v vs. viii in Figs. 6A–6C).

18α-GA-treated MLO-Y4 osteocytes with a loading history in the present study reflected a similar condition to the conditional deletion of Cx43 in mice, since osteocytes in vivo are always under a complicated loading environment. Therefore, the conditional deletion of Cx43 in mice is likely to block the gap junction in MLO-Y4 osteocytes with a certain loading history but not in those without any history of mechanical stimuli exposure. Therefore, our in vitro findings are consistent with those of previous reports in vivo.

A previous study showed a greater response to loading at the endocortical surface than the periosteal surface (Birkhold et al., 2016). However, the finite elements analysis in this above-mentioned study (Birkhold et al., 2016) showed that the endocortical surface was less strained than the periosteal surface. Our findings may provide a possible explanation for this observation that osteocytes under higher-magnitude loading may be more habituated than those under lower-magnitude loading, thereby showing less sensitivity to mechanical stimuli. In addition, we found that the expression of our interesting genes in MLO-Y4 osteocytes under 31 dynes/cm2 CCF was more sensitive to additional CCF than MLO-Y4 osteocytes that had previously been exposed to 63 dynes/cm2 CCF (Figs. 6 and 8).

A number of previous studies have reported an enhanced response to mechanical stimulation in mice with conditional deletion of Cx43 from either osteoblasts or osteocytes (Zhang et al., 2011), osteochondroprogenitors (Grimston et al., 2012), and osteocytes only (Bivi et al., 2013). These reports supported our present finding that 18α-GA treatment increased the sensitivity to CCF in the MLO-Y4 osteocytes with a loading history (Fig. 6). Deletion of Cx43 in mature osteoblasts and osteocytes (Plotkin et al., 2008) or in osteocytes only (Bivi et al., 2012) does not decrease the bone mass. Quite the contrary, it increases the periosteal bone formation in the pattern of a bone subjected to modeling during growth (Bivi et al., 2012), which is in agreement with our results that blockade of the gap junction by 18α-GA decreased the RANKL/OPG ratio.

Given the above, our findings suggest that gap junctional intercellular communications may be involved in the habituation of osteocytes.

The mRNA expression of OPG, Gja1 and Dmp1 showed significant correlation to cell morphological changes during the recovery period

Intriguingly, the expression profile of OPG, Gja1 and Dmp1 showed a significant correlation with the morphological changes in the MLO-Y4 osteocytes with a loading history (Figs. 5I–5L). Gja1/Cx43 has been reported to control cell morphology and migration in other types of cells (Xu et al., 2006; Liu et al., 2012; Machtaler et al., 2014). However, the authors do not find the previous studies reported the correlations between cell morphological changes and the mRNA expression of OPG or Dmp1.

Notably, both the solidity and cell area of the MLO-Y4 osteocytes with a loading history were significantly different from in those without a loading history after the 10-h recovery period, even it was at the same level at the beginning of the recovery period (Figs. 4C and 4D). Similarly, more significant changes of all of the mRNA expression profile of Dmp1 (Fig. 5F) were observed at the end of the 10-h recovery period, which is significantly correlated to the solidity and cell area (Figs. 5K and 5L). According to the definition of the solidity in the ImageJ/Fiji software program (Schindelin et al., 2012), a lower solidity may suggest more cellular dendritic processes in MLO-Y4 osteocytes since the expended slim dendritic processes would greatly increase the convex area but contribute slightly to the entire cell area (Fig. 2C). It was also demonstrated that fluid shear stress changed the cell morphology of osteoblast-like IDG-SW3 cells (Xu et al., 2018), and 48 h of clinorotation changed the α-tubulin distribution in MLO-Y4 cells (Xu et al., 2012). Indeed, our results showed that the cell morphology was not only immediately changed after the mechanical stimuli exposure but also gradually changed over 10 h after a long period of CCF loading (Fig. 4).

A previous study from our group showed that mechanical loading induced morphological changes in osteocytes in vivo (Sugawara et al., 2013). Recent studies showed that the cell morphological changes determine the direction of human mesenchymal stem cell (hMSC) differentiation through activating or inhibiting differential biological signals (Fan et al., 2019). For example, hMSCs with a spread shape will differentiate to osteoblasts, those with a round shape will differentiate to adipocytes, etc. Actin, one of the three major components of the cytoskeleton, regulates the cell shape by controlling the dynamic equilibrium between the monomeric and filamentous states, which also presents in the nucleus for transcriptional regulation (Misu, Takebayashi & Miyamoto, 2017). There are numerous studies showing that morphological differences in osteocytes always correlate to some changes in gene expression or metabolomics (Bacabac et al., 2008; Sasaki et al., 2012; Xu et al., 2012); however, the underlying mechanism remains unclear and should be examined in future studies.

Given the discussion above, our findings suggest that CCF-induced cell morphological changes may be involved in the habituation of osteocytes.

Limitations

The approach reported in this study is economic and convenient; however, the limitations of this approach are also remarkable. First, the CCF was not delivered to every cell since the area of the cover glass is smaller than the area of cell culture dish or well (Figs. S1 and S2). Second, the tolerance in the size of the cover glass may cause a variance in the CCF strength. These above-mentioned limitations may generate a considerable variance, which can be noticed from all our results. For example, since the linear regression test was performed using the individual mRNA expression fold change against the mean value of morphological changes, the discrepancy between P-values for Pearson’s correlation coefficient and non-zero slope in Fig. 5J suggested a high intraclass variance, which was also suggested by the long error bars in Figs. 5A–5H and 6.

Summary

Our findings showed that the mRNA expression of RANKL, OPG, Sost and OCN in MLO-Y4 osteocyte-like cells with treatment of loading history showed the decrement in responsiveness to a re-application of the same CCF as the loading history (Fig. 6). This habituation phenomenon could be disrupted by a GJIC inhibitor, which suggested the MLO-Y4 osteocyte-like cells network is important for this habituation in vitro (Fig. 6). This behavior of in vitro cultured MLO-Y4 osteocyte-like cells network is pretty similar to that of mathematical modeling of small world networks (Strogatz, 2001) that was used to explain how rest-inserted loading arises more bone formation than continuous cyclic loading (Gross et al., 2004). A small world network exhibit power law behavior, in which small magnitude events occur often, whereas large magnitude events occur infrequently (Latora & Marchiori, 2001; Strogatz, 2001; Gross et al., 2004). A small world network also exhibits threshold behavior (Latora & Marchiori, 2001; Strogatz, 2001; Gross et al., 2004). The decreased GJIC by the treatment of loading history (Fig. 7) blocked the connections between the MLO-Y4 osteocyte-like cells then consequently decreased the efficient of the small world network since the decreased connection range (k) and probability (p) in this small world mathematic model (Latora & Marchiori, 2001). Therefore, the raised threshold of the small word network by efficient changes may be a possible mechanism to explain the habituation behavior of in vitro cultured MLO-Y4 osteocyte-like cells observed in this study. However, this hypothesis needs to be further confirmed by more supportive evidence in the future. Since the osteocytes network is important for bone adaptation (Robling, 2012), findings about the behavior of in vitro cultured MLO-Y4 osteocyte-like cells network in response to loading history from this study may have contributions to the further understanding in the molecular mechanism of the bone adaptation and finally may contribute to reduce the risk of bone fracture by mediation of bone adaptation.

Supplemental Information

Supplemental Information 1 Assignment of treatment for 24-well plate.

Experiment design for RNA extraction and cell morphological measurements.

Click here for additional data file.

Supplemental Information 2 An example view at the edge of the cover glass.

After 46 h of exposure to 31 dynes/cm2 CCF (MLO-Y4 cells without a loading history). The cover glass was removed when starting the Calcein-AM and Hoechst 33342 staining.

Click here for additional data file.

Ziyi Wang would also like to thank Shanqi Fu and Dr. Takako Hattori of the Department of Biochemistry and Molecular Dentistry of Okayama University Graduate School of Medicine Dentistry and Pharmaceutical Sciences for providing valuable advice.

Additional Information and Declarations

Competing Interests

Author Contributions

Data Availability

The authors declare that they have no competing interests.

Ziyi Wang conceived and designed the experiments, performed the experiments, analyzed the data, prepared figures and/or tables, authored or reviewed drafts of the paper, funding acquisition and providing resources, and approved the final draft.

Yao Weng performed the experiments, analyzed the data, prepared figures and/or tables, authored or reviewed drafts of the paper, and approved the final draft.

Yoshihito Ishihara conceived and designed the experiments, analyzed the data, authored or reviewed drafts of the paper, and approved the final draft.

Naoya Odagaki conceived and designed the experiments, analyzed the data, authored or reviewed drafts of the paper, and approved the final draft.

Ei Ei Hsu Hlaing conceived and designed the experiments, analyzed the data, authored or reviewed drafts of the paper, and approved the final draft.

Takashi Izawa analyzed the data, authored or reviewed drafts of the paper, funding acquisition and providing resources, and approved the final draft.

Hirohiko Okamura analyzed the data, authored or reviewed drafts of the paper, supervision, and approved the final draft.

Hiroshi Kamioka conceived and designed the experiments, performed the experiments, analyzed the data, authored or reviewed drafts of the paper, supervision, and approved the final draft.

The following information was supplied regarding data availability:

An example view at the edge of the cover glass is available as a Supplemental Figure.

All the raw data is available at Mendeley Data: Wang, Ziyi; Weng, Yao; Ishihara, Yoshihito; Kamioka, Hiroshi (2020), “Loading history changes the morphology and compressive force-induced expression of receptor activator of nuclear factor kappa B ligand/osteoprotegerin in MLO-Y4 osteocytes”, Mendeley Data, v1 DOI 10.17632/2yfd2w8jfp.1

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
