# Peer review of "Loading history changes the morphology and compressive force-induced expression of receptor activator of nuclear factor kappa B ligand/osteoprotegerin in MLO-Y4 osteocytes"

_PeerJ, doi:10.7717/peerj.10244_

## Round 0.1 · original submission · Major Revisions

All reviewers have identified major issues with the described research. Reviewers 1 and 3 in particular identified specific points that have to be addressed with revisions of the text and/or additional experimental work.
For example, there is concern about the phenotype of the cell line. It has to be demonstrated that the cells used in the study are able to differentiate and function properly. Especially, it has to be established that these cellular properties are maintained at the late passages used for evaluation.

Based on the reviewers' comments, I encourage you to revise your manuscripts and respond to each of the individual points raised in the reviews.

Reviewer 1 ·

Basic reporting

1) This reviewer suggests authors to cite the references appropriately. Please add missing references. Add appropriate reference for lines 219, 227-228, 230 and 258.

2) General comment for all figures: please provide more experimental details and expand the figure legends and describe x and y axis.

Experimental design

Line 143. Authors included a general statement that all experiments were performed in triplicates. However throughout the results, it is not clear whether the experiment were performed independently three times and under identical conditions. Please describe in detail about the experimental conditions, set-up format and number of independent experiments for figure 3-6.

Validity of the findings

1) Line 155. Please describe what was the criteria for selecting 31 and 63 CCF for studying osteocytes. Additionally, what is the physiological relevance for such CCF compression forces for osteocytes. Also, please discuss the overall implication of this study in context to both normal physiology and diseased conditions.

2) Line 234. Authors did not reported such analysis in the manuscript. Please explain.

3) Figure 2. There is a discrepancy for reported time points between cells exposed to 31 CCF (36, 39, 42 and 46h) and 63 CCF (36 h). Currently, results described with Figure 2A-B are misleading and not very clear.

Additional comments

1) Figure 1. It would be beneficial for the readers if authors can provide the overview of the experimental design in figure legends.
2) Figure 1. scale bar is not visible.
3) Figure 2A scale bar is missing
4) Figure 2B scale bar is not visible
5) Figure 2A-B. Overall, the resolution of figures is very poor.

Reviewer 2 ·

Basic reporting

no comments

Experimental design

no comments

Validity of the findings

no comments

Additional comments

1. Introduction section, first paragraph, first sentence does not make sense, rewrite it.
2. Application of CCF force magnitude and duration was determined based on which in vivo situation in human such as regular exercise, walking, running……and how was that determined?
3. What is the translational value of the findings of this study (in vivo or in preclinical studies)?

·

Basic reporting

The phenotypes of cell line were insufficiently characterized. It was unclear if the cells were adequately differentiated and functional, especial that they were assayed at 39 passages.

Several statements were insufficiently supported by references or explanations see the minor comments.

1) Introduction, p7, lines 55-56: Add references to support “Therefore, the bone cell network is considered to be encoded with a unique system for maintaining the history of mechanical stimuli received.”

2) Introduction, p7 lines 64-65: Add references to support “Therefore, bone cells may store and modify long-term mechanical memory through structural changes in the lacuna-canalicular system.”

3) Introduction, p8 lines 70-71: Add references to support “Habituation is a term that describes the decrement in responsiveness to a repetitive stimulus in neuronal systems.”

4) Introduction, p8 lines 75-76: Add references to support “The relative amount and distribution of RANKL and OPG proteins is thought to determine the destination of osteoclasts.”

5) Introduction, p8 lines 79-80: Delete or add references in “However, very few studies have investigated the habituation of osteocytes to mechanical loading using an in vitro culture system.”

6) Blockade of junctions partially rescued the habituation effect, line 219, p15: Move into the result section to explain the rationale of 18α-GA treatment “18α-GA could block the gap junctions”

Experimental design

Cell culture and reagents, p9 lines 87-94: There were no information how phenotypes of MLO-Y4 cell line was checked and if they acted as osteocytes, especially that they were selected at 39 passages in “The MLO-Y4 cells line was purchased from Kerafast (Boston, MA, USA). MLO-Y4 cells (at 39 passages), an osteocyte-like cell line derived from long bone of a transgenic female mouse containing the osteocalcin promoter driving SV40 T-antigen (Kato et al., 1997), were seeded onto type I collagen-coated 24-well plates and cultured in alpha-modified Eagle's minimal essential medium (α-MEM; Thermo Fisher Scientific, Waltham, MA, USA) containing 5.0% heat-inactivated fetal bovine serum (HIFBS; HyClone Laboratories, Logan, UT, USA), 5.0% fetal calf serum (FCS; HyClone), 100 U/ml penicillin, and 100 mg/ml streptomycin (Thermo Fisher Scientific) at 37 °C with 5% CO2.”

Validity of the findings

The statistical analysis was not adequately supported by independent experiments to validate the findings.

1) Statistical analyses, line 143, p 11: It was unclear if there were independent measurements to obtain reliable statistics. Triplicates obtained from the same sample are insufficient to obtain reliable statistics in “All experiments in this study were performed in triplicate.”

There were a lack of experiments to support hypothesis as indicated:
1) In vitro cultured MLO-Y4 osteocytes were successfully habituated, line 210-211, p15: Delete since there were no findings associated with plasma membrane disruption or perform experiments to support “The habituation phenomena that we observed in the present study may be related to plasma membrane disruption (PMD) and PMD repair under conditions of heavy mechanical loading.”

2) Blockade of junctions partially rescued the habituation effect, line 219, p15: Move into the result section to explain the rationale of 18α-GA treatment “18α-GA could block the gap junctions”

3) Blockade of junctions partially rescued the habituation effect, line 219 p15 and lines 228-230 p16: Additional findings are needed to support adequately “Connexins are main components of gap junctions and hemichannels, and the most abundant connexin in bone is connexin43 (Cx43), which is also expressed by osteocytes (Ishihara et al., 2008).” And “Therefore, the conditional deletion of Cx43 in mice is likely to block the gap junction in MLO-Y4 osteocytes with a certain loading history but not in those without any history of mechanical stimuli exposure”

4) The expression OPG showed significant correlation to cell morphological changes during recovery period, lines 273-275, p18: Additional experiments are needed to support “Therefore, it is reasonable to hypothesize that mechanical stimuli-induced morphological changes may influence the gene expression profile in osteocytes for a considerably long time.”

Additional comments

Minor comments :
1) Abstract, p1: Specify OPG in ” The RANKL/OPG expression, cell numbers, viability, and morphology were time-dependently examined at 0, 3, 6, and 10 h.”


2) The morphology changed during the recovery period lines 161-162, p12: Specify aspect ratio in “Figure 2C shows an example of the morphological measurements. After changing the CCF from 63 dynes/cm2 back to 31 dynes/cm2 for 10 h (recovery period), the aspect ratio was significantly increased only in the MLO-Y4 osteocytes without a loading history (Fig. 4A).”

3) Conclusion, lines 309-314, p20: Delete conclusion since it was repeated in the discussions “In conclusion, in order to investigate the influence of the loading history on the RANKL/OPG expression and morphology of MLO-Y4 osteocytes, we established a useful in vitro model using osteocyte-like MLO-Y4 cells. In this study, the in vitro cultured MLO-Y4 osteocytes showed habituation of RANKL/OPG expression in their response to long-duration (36 h) CCF. Gap junctional intercellular communication and cell morphology may play a role in this habituation phenomena in MLO-Y4 osteocytes.”

---

## Round 0.2 · accepted · Accept

Dear Authors:

Both reviewers are satisfied with the revisions you have made. Your manuscript will now be processed for publication. Thank you for your contribution!

With best wishes - Ursula Stochaj

Reviewer 1 ·

Basic reporting

no comments

Experimental design

no comments

Validity of the findings

no comments

Additional comments

Revised manuscript has incorporated all the suggestions and I do not have any additional comments.

Reviewer 2 ·

Basic reporting

N/A

Experimental design

N/A

Validity of the findings

N/A

Additional comments

Current version is good enough to publish.